# Structured Physical Attributes Enable Efficient Foundation Models for Land-Surface Prediction

**Anonymous Authors[1]**

## Abstract

Foundation models for structured tabular and time-series data have demonstrated strong zero-shot and transfer capabilities across general-purpose prediction tasks, yet their applicability to domain-governed physical systems remains underexplored. We present StefaLand, a structured foundation model pretrained on tabular landscape attributes and meteorological time-series across 8,634 global catchments, targeting two core Earth system prediction tasks: streamflow forecasting and soil moisture estimation. StefaLand employs a transformer-based masked autoencoder with cross-variable group masking to learn interactions among physically related attribute groups, and is adapted to downstream tasks through lightweight residual adapters. Against strong baselines including TabPFN, TiRex, AlphaEarth embeddings, and supervised LSTM models, StefaLand achieves consistent gains in spatial generalization under strict holdout regimes, reducing streamflow RMSE by approximately 20% over supervised baselines in ungauged basin experiments. Our results demonstrate that domain-specialized pretraining on structured physical attributes provides stronger cross-domain transfer than general-purpose tabular foundation models, and offers a computationally accessible alternative to large-scale vision-based Earth observation models.

## 1. Introduction

Climate change is intensifying the frequency and severity of geohazards (Ebi et al., 2021; IPCC, 2021), creating urgent demand for models that accurately predict land-surface dynamics. Here we focus on two societally critical targets: streamflow, where prediction failures cause flooding

[1]Anonymous Institution, Anonymous City, Anonymous Region, Anonymous Country. Correspondence to: Anonymous Author <anon.email@domain.com>.

Preliminary work. Under review by the International Conference on Machine Learning (ICML). Do not distribute.

and hydrologic drought, and soil moisture, which controls ecosystem health and land-atmosphere interactions (Dorigo et al., 2013a).

A key challenge for geoscientific ML is spatial generalization: observational data are sparse and spatially imbalanced, with gauge networks dense in developed nations but scarce across Africa, South America, and much of Asia. Models trained on data-rich regions often degrade substantially when applied elsewhere (Feng et al., 2023), yet most geoscientific ML models remain supervised approaches trained for narrow tasks. Existing vision-based foundation models focus on surface appearance and spatial patterns but have not targeted the cross-domain interactions among landscape attributes (climate, soil, vegetation, terrain, and geology) that govern dynamic land-surface processes (Xie et al., 2023), and to our knowledge no foundation model has been developed with a primary focus on dynamical land-surface modeling.

Supervised LSTMs remain the dominant and difficult-to-surpass baseline for geophysical time series, with transformers generally failing to outperform them on continuous regression signals (Zeng et al., 2022; Liu et al., 2025). Recent geoscientific foundation models including TerraMind (Jakubik et al., 2025), Aurora (Bodnar et al., 2025), AlphaEarth (Brown et al., 2025), and Galileo (Tseng et al., 2025) are pretrained on remote sensing imagery and transfer well as static spatial descriptors; AlphaEarth in particular provides strong globally available embeddings that serve as a competitive baseline for spatial generalization. On the structured data side, TabPFN (Hollmann et al., 2025) and TiRex (Auer et al., 2025) have demonstrated strong zero-shot transfer on tabular and time-series benchmarks, but their applicability to physically governed systems with strong spatial structure remains untested.

StefaLand (Spatial-Temporal Earth Foundation model with Attributes for the Land Surface) addresses this gap directly, pretraining on tabular landscape attributes and meteorological time series across 8,634 global catchments. Its core design combines a masked autoencoder backbone with cross-variable group masking over physically related attribute groups, encouraging the model to learn cross-domain interactions that neither general-purpose structured FMs nor

*Figure 1.* Conceptual overview of the StefaLand Structure. Static landscape attributes and dynamic forcings are jointly embedded using a transformer-based masked autoencoder with cross-variable group masking.

vision-based EO models are designed to capture. Under strict spatial holdout regimes, StefaLand achieves strong generalization on streamflow and soil moisture, outperforming TabPFN, TiRex, supervised LSTM baselines, and large-scale vision-based EO embeddings. This is achieved with a computationally accessible design, roughly 12 million parameters pretrained in approximately 720 V100 GPU hours, offering a practical alternative to vision-based Earth observation models trained on orders of magnitude more data.

## 2. Methods

### 2.1. StefaLand Structure

StefaLand is a transformer-based masked autoencoder inspired by bidirectional language models such as BERT (Devlin et al., 2018), designed to jointly embed static landscape attributes and dynamic time-varying forcings within a unified representation.

During pretraining, StefaLand uses cross-variable group masking, in which physically or statistically related variables are masked together rather than independently. This prevents trivial reconstruction through correlated inputs and encourages the model to capture cross-domain linkages, such as interactions between soil texture and climate seasonality or between topography and hydrologic response. Grouped masking strategies have previously been explored in multimodal learning settings, including Presto (Tseng et al., 2023), where groupings have typically followed sensor or modality boundaries. Here, groupings are defined by physical inter-domain relationships rather than data source, a notable distinction that forces the model to internalize the coevolution of landscape attributes rather than simply reconstructing within-sensor correlations.

StefaLand is pretrained on a global dataset spanning 8,634 catchments over 40 years; full variable lists and group assignments are in Appendix C.

### 2.2. Finetuning for Prediction Tasks

We adapt the pretrained StefaLand encoder to downstream prediction tasks using a lightweight adapter **StefaLand-resConn**, it integrates pretrained embeddings with raw meteorological forcings through a residual adaptation pathway via additive fusion prior to the recurrent decoder. Let $E_t$ denote the StefaLand embedding at time $t$, and let $x_t$ denote the corresponding raw forcing inputs. We compute:

$$r_t = f_{\text{conv+linear}}(x_t), \tag{1}$$
$$h_t = \text{LSTM}(E_t + r_t), \tag{2}$$
$$\hat{y}_t = W_o h_t + b_o, \tag{3}$$

where $f_{\text{conv+linear}}(\cdot)$ denotes a shallow convolutional and linear projection block. Residual connections propagate both the pretrained embedding $E_t$ and the task-specific signal $r_t$ into the recurrent decoder, strengthening spatial generalization while adapting to local dynamics.

## 3. Experiments

Hyperparameters were tuned with Ray Tune and kept consistent across model configurations within each experimental case, using temporal validation splits to evaluate spatial generalization. Full details are in Appendix C.

**Model variants and baselines.** Supervised baselines include LSTM-SL, the dominant architecture for streamflow prediction (Kratzert et al., 2019; Feng et al., 2021), and representative transformer-based sequence models: Informer (Zhou et al., 2021), Reformer (Kitaev et al., 2020), and DLinear (Zeng et al., 2022). For pretrained Earth representations we include AlphaEarth LSTM and AlphaEarth ResConn (Brown et al., 2025), which feed spatially aggregated AlphaEarth embeddings into an LSTM decoder with and without a residual adapter. We additionally evaluate TabPFN (Hollmann et al., 2025), provided with compact

statistical summaries of each historical window to accommodate its fixed context window, and TiRex (Auer et al., 2025) as a time-series foundation model baseline.

### 3.1. CAMELS Streamflow Prediction

To compare spatial generalization on a well benchmarked dataset, we follow (Feng et al., 2021), testing prediction in ungauged basins (PUB, random spatial K-fold) and ungauged regions (PUR, regional spatial K-fold). We use CAMELS (Addor et al., 2017; Newman et al., 2014), restricted to the 531-basin subset with clear watershed boundaries (Newman et al., 2017). Basins were divided into 10 random groups for PUB and 7 contiguous regions for PUR, employing leave-one-out in both cases. To avoid leakage, all CAMELS-overlapping stations were removed during pretraining for PUB, and entire regions were excluded for PUR.

*Table 1.* CAMELS streamflow PUB results (random spatial holdout). Median values, ± is standard error across folds.

| Model | RMSE ↓ | ubRMSE ↓ | Corr ↑ | NSE ↑ |
|---|---|---|---|---|
| LSTM SL | 1.402 ± 0.04 | 1.360 ± 0.04 | 0.762 ± 0.01 | 0.636 ± 0.04 |
| DLinear | 2.012 ± 0.06 | 2.000 ± 0.06 | 0.598 ± 0.04 | 0.302 ± 0.48 |
| Informer | 2.262 ± 0.07 | 2.237 ± 0.08 | 0.521 ± 0.01 | 0.104 ± 0.08 |
| Reformer | 1.908 ± 0.09 | 1.871 ± 0.09 | 0.718 ± 0.00 | 0.270 ± 0.18 |
| TiRex | 3.339 ± 0.41 | 3.012 ± 0.38 | 0.583 ± 0.02 | 0.250 ± 0.20 |
| TabPFN | 2.727 ± 0.09 | 2.725 ± 0.09 | 0.718 ± 0.01 | 0.509 ± 0.14 |
| AlphaEarth LSTM | 1.409 ± 0.10 | 1.393 ± 0.10 | 0.837 ± 0.11 | 0.618 ± 0.05 |
| AlphaEarth resConn | 1.361 ± 0.09 | 1.345 ± 0.09 | 0.859 ± 0.01 | 0.647 ± 0.04 |
| **StefaLand resConn** | **1.111 ± 0.04** | **1.068 ± 0.04** | **0.869 ± 0.01** | **0.717 ± 0.16** |

*Table 2.* CAMELS streamflow PUR results (regional spatial holdout). Median values, ± is standard error across folds.

| Model | RMSE ↓ | ubRMSE ↓ | Corr ↑ | NSE ↑ |
|---|---|---|---|---|
| LSTM SL | 1.609 ± 0.24 | 1.457 ± 0.22 | 0.743 ± 0.02 | 0.554 ± 0.13 |
| DLinear | 2.019 ± 0.35 | 1.983 ± 0.34 | 0.597 ± 0.03 | 0.290 ± 0.99 |
| Informer | 2.332 ± 0.41 | 2.295 ± 0.37 | 0.497 ± 0.01 | 0.046 ± 0.27 |
| Reformer | 2.074 ± 0.33 | 1.978 ± 0.31 | 0.686 ± 0.02 | 0.257 ± 1.17 |
| TiRex | 3.475 ± 0.97 | 3.236 ± 0.76 | 0.434 ± 0.08 | 0.187 ± 0.28 |
| TabPFN | 2.709 ± 1.08 | 2.501 ± 0.89 | 0.576 ± 0.11 | 0.328 ± 0.12 |
| AlphaEarth LSTM | 1.724 ± 0.85 | 1.685 ± 0.82 | 0.780 ± 0.07 | 0.456 ± 0.15 |
| AlphaEarth resConn | 1.727 ± 0.74 | 1.684 ± 0.71 | 0.790 ± 0.06 | 0.520 ± 0.10 |
| **StefaLand resConn** | **1.344 ± 0.21** | **1.334 ± 0.19** | **0.801 ± 0.02** | **0.635 ± 0.25** |

StefaLand-resConn achieves the strongest performance across all metrics in both PUB and PUR evaluation (Tables 1 and 2), reducing RMSE by approximately 20% under PUB and 16–17% under PUR relative to the supervised LSTM baseline. Both general-purpose structured FMs perform poorly: TiRex yields the highest RMSE of all models despite being a purpose-built time-series FM, and TabPFN fails to capture key temporal structure in rainfall-runoff dynamics despite moderate correlation, suggesting that domain-specialized pretraining on physically structured attributes provides transfer advantages that general-purpose structured FMs, pretrained without physical inductive biases, are not capturing. AlphaEarth-based variants improve over LSTM particularly in correlation, but remain well below StefaLand-resConn even with residual feature reuse. The gains demonstrated are also substantial: to provide some context, when

LSTM raised NSE from 0.64 to 0.73 in the *temporal* case (without ensemble), it was considered a generational change in predictive performance (Nearing et al., 2021; Feng et al., 2021).

### 3.2. Global Soil Moisture

We evaluated finetuning StefaLand for soil moisture predictions following (Liu et al., 2023a), using ISMN (Dorigo et al., 2011; 2013a). ISMN consists of 1,316 ground-based stations. We performed five-fold spatial cross-validation for random holdout and a regional holdout on Europe, training on all other continents while excluding European sites (129) for testing. LSTM again serves as the established state-of-the-art baseline (Wang et al., 2024; Liu et al., 2023b).

*Table 3.* Soil moisture prediction across ISMN (random location holdout). Values are median ± standard error across folds.

| Model | RMSE ↓ | ubRMSE ↓ | Corr ↑ |
|---|---|---|---|
| LSTM SL | 0.073 ± 0.002 | 0.055 ± 0.001 | 0.764 ± 0.006 |
| DLinear | 0.088 ± 0.001 | 0.065 ± 0.001 | 0.612 ± 0.007 |
| Informer | 0.102 ± 0.002 | 0.082 ± 0.001 | 0.232 ± 0.012 |
| Reformer | 0.090 ± 0.002 | 0.071 ± 0.001 | 0.568 ± 0.015 |
| TiRex | 0.084 ± 0.007 | 0.069 ± 0.005 | 0.535 ± 0.011 |
| TabPFN | **0.068 ± 0.004** | 0.057 ± 0.003 | 0.404 ± 0.016 |
| AlphaEarth LSTM | 0.075 ± 0.001 | 0.062 ± 0.001 | 0.427 ± 0.019 |
| AlphaEarth resConn | 0.082 ± 0.001 | 0.067 ± 0.001 | 0.406 ± 0.012 |
| **StefaLand resConn** | **0.068 ± 0.001** | **0.054 ± 0.001** | **0.783 ± 0.005** |

*Table 4.* Soil moisture prediction across ISMN cross-continental validation on Europe (129 sites).

| Model | RMSE ↓ | ubRMSE ↓ | Corr ↑ |
|---|---|---|---|
| LSTM SL | 0.112 | 0.053 | 0.510 |
| DLinear | 0.093 | **0.051** | 0.623 |
| Informer | 0.088 | 0.063 | 0.358 |
| Reformer | 0.087 | 0.058 | 0.553 |
| TiRex | 0.093 | 0.071 | 0.423 |
| TabPFN | **0.081** | 0.054 | 0.401 |
| AlphaEarth LSTM | 0.087 | 0.068 | 0.313 |
| AlphaEarth resConn | 0.090 | 0.071 | 0.308 |
| **StefaLand resConn** | 0.090 | 0.059 | **0.638** |

StefaLand-resConn achieves the strongest correlation in both evaluation settings (Tables 3 and 4): 0.783 under random holdout with lowest ubRMSE (0.054) and best RMSE (0.068), and 0.638 under cross-continental holdout on Europe, exceeding all baselines in both cases. Although TabPFN matches RMSE under random holdout, its substantially lower correlation indicates weaker agreement with temporal variability. AlphaEarth-based variants show limited transfer despite similar error magnitudes, suggesting that geoscience-pretrained structured representations are critical for robust generalization under spatial distribution shift.

### 3.3. Ablations

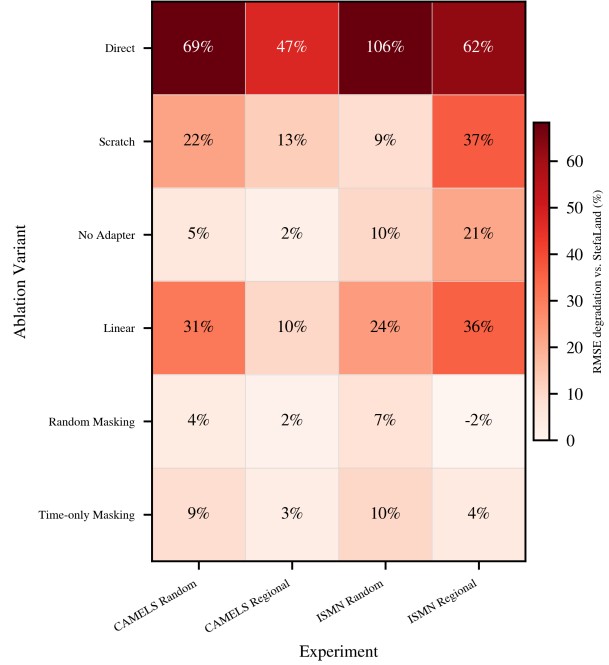

*Figure 2.* Ablation impact matrix across evaluation settings using RMSE. Each cell shows the percent RMSE increase relative to the full StefaLand ResConn model (lower is better)

To isolate the contributions of pretraining, task adaptation, and cross-variable group masking, we compare six variants against the full model: direct prediction (no pretraining), scratch training (full architecture, random initialization), no adapter, linear probing, random masking (variables masked independently without physical grouping), and time-only masking (masking over temporal windows only). All masking ablations retain the full StefaLand architecture. As shown in Figure 2, degradation is dominated by the removal of pretraining, with direct prediction and scratch training yielding the largest errors especially under spatial generalization. Adapter choice is secondary but consistent. Both masking ablations show measurable degradation across most settings, confirming that physically motivated cross-variable grouping contributes to cross-domain representation learning. Notably, ubRMSE and correlation favor CVGM even where raw RMSE differences are small, suggesting this grouping improves temporal agreement beyond error magnitude alone. Full numerical results are provided in Appendix B.3.

## 4. Discussion

Methods that reliably improve spatial generalization to data-scarce regions remain rare in the literature (Beery et al., 2018; Gacu et al., 2025). StefaLand combined with lightweight task-specific heads achieves state-of-the-art or competitive performance across tasks examined, with the strongest gains arising from architectures that combine pretrained representations with explicit temporal modeling via residual connections. This indicates that attribute-centric pretraining captures problem-relevant structure while downstream sequence models resolve task-specific dynamics, and together these results support the premise that foundation models can improve out-of-domain transfer and help democratize prediction quality in data-scarce regions.

Across evaluation settings, a consistent pattern emerges: pretraining on landscape attributes yields representations that are highly relevant to hydrologic and land-surface prediction tasks, enabling stronger spatial generalization than purely supervised approaches or training from scratch. Comparisons with alternative pretrained representations further suggest that problem relevance of the pretraining signal is at least as important as scale alone, particularly for tasks governed by physical and environmental processes rather than visual appearance. General-purpose structured FMs such as TiRex and TabPFN, pretrained without physical inductive biases, underperform even supervised baselines here, reinforcing that benchmark-oriented pretraining does not readily transfer to physically governed systems with strong spatial autocorrelation. This attribute-based approach is therefore complementary to both general-purpose structured FMs and satellite-centric foundation models: the former excel on i.i.d. benchmarks but lack physical inductive biases, while the latter extract large-scale visual patterns but do not directly encode the structured environmental attributes that govern land-surface processes. StefaLand occupies a distinct niche between these two families, offering targeted physical representations without requiring massive image archives or domain-agnostic pretraining corpora.

Several limitations remain. The selection of geological- and ecologically-focused attributes could be expanded to better characterize the subsurface, and two-dimensional data such as elevation maps could be selectively incorporated using vision transformer heads in the future. Expanding the range of prediction targets to include evapotranspiration, snow water equivalent, and groundwater levels would further broaden applicability. Overall, attribute-centric pretraining combined with lightweight temporal heads delivers strong spatial generalization while remaining computationally accessible, pointing toward a practical path for high-quality predictions in regions where they are most needed but data are most limited.

## 5. Reproducibility

The model files and all code is released publicly at [https://anonymous.4open.science/r/StefaLandRelease-3731/]. All datasets used are public. A complete list of variables used for each task, along sources, is in Appendix C.

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

## A. Detailed Model Architecture

This appendix provides the complete mathematical formulation of the StefaLand model architecture.

### A.1. Embedding Dynamic and Static Inputs

StefaLand independently embeds each dynamic and static variable into a latent space. Specifically, for each dynamic variable $c$ at each time step $t$, a two-step nonlinear embedding is applied individually:

$$z_{t,c} = \text{GELU}(x_{t,c}W_{1,c} + b_{1,c})W_{2,c} + b_{2,c} \tag{4}$$

where $W_{1,c} \in \mathbb{R}^{1 \times 64}$ and $W_{2,c} \in \mathbb{R}^{64 \times 256}$ are embedding parameters. After embedding all dynamic variables individually, embeddings are stacked and summed across the variable dimension, resulting in a single embedding vector per time step:

$$z_t = \sum_{c=1}^{C} z_{t,c} \tag{5}$$

Similarly, static attributes are embedded individually:

$$z_{\text{static},i} = \text{GELU}(s_i W_{1,i} + b_{1,i})W_{2,i} + b_{2,i} \tag{6}$$

where separate embedding layers are used for static features. These individual static embeddings are then concatenated with dynamic embeddings along the temporal dimension, resulting in a unified embedding tensor:

$$Z = [z_1; z_2; \ldots; z_T; z_{\text{static}}] \tag{7}$$

This static embedding acts as a global learnable token, allowing the model to incorporate basin-specific context into temporal dynamics at any depth of the Transformer layers.

### A.2. Location-aware Cross-Variable Group Masking

StefaLand introduces Cross-Variable Group Masking (CVGM), a masking strategy that forces the model to capture interactions among correlated hydrologic variables rather than treating them independently. Given a predefinition of hydrological variables into groups $G = \{g_1, g_2, \ldots, g_k\}$, masking occurs as follows:

1. A temporal masking window $[\tau, \tau + \ell)$ is randomly sampled with length $\ell \sim \mathcal{U}(L_{\min}, L_{\max})$.

2. For each variable group $g_k$, a Bernoulli mask indicator $m_k \sim \text{Bernoulli}(p_{\text{mask}})$ determines whether the group is masked.

3. For each time step $t$ within the masked temporal window and each variable $c$ belonging to masked groups, the embedded feature vector is replaced by a learned mask vector $\mathbf{m}_c$.

Each hydrologic variable $c$ has its own trainable mask embedding vector $\mathbf{m}_c \in \mathbb{R}^{256}$. This CVGM procedure creates reconstruction targets that require modeling cross-variable dependencies and physical interactions.

### A.3. Learnable Positional Encoding

To provide positional information, StefaLandemploys learnable positional encoding. Each position $i$, corresponding to each time step and the appended static embedding, is assigned a trainable embedding vector $\mathbf{p}_i$. The encoded embedding becomes:

$$\tilde{Z} = Z + P \tag{8}$$

where $P = [\mathbf{p}_1; \ldots; \mathbf{p}_{T+1}]$.

### A.4. Transformer Encoder

The embeddings enriched by positional encoding are processed through an $N$-layer Transformer encoder, where each Transformer block successively applies Multi-Head Self-Attention (MHA) with $h$ attention heads, followed by a residual connection and Layer Normalization. Subsequently, a position-wise Feedforward Network (FFN) is applied, also followed by another residual connection and Layer Normalization:

$$A^{(\ell)} = \text{MHA}(H^{(\ell-1)}) \tag{9}$$

$$\tilde{H}^{(\ell)} = \text{LayerNorm}(H^{(\ell-1)} + A^{(\ell)}) \tag{10}$$

$$F^{(\ell)} = \text{FFN}(\tilde{H}^{(\ell)}) \tag{11}$$

$$H^{(\ell)} = \text{LayerNorm}(\tilde{H}^{(\ell)} + F^{(\ell)}) \tag{12}$$

### A.5. Reconstruction of Original Inputs

The final hidden states from the Transformer encoder, $H^{(N)}$, are linearly projected and passed through a single-layer bidirectional LSTM to capture the local temporal dependencies and continuity:

$$U = \text{LSTM}(H^{(N)} W_{\text{enc-proj}} + b_{\text{enc-proj}}) \tag{13}$$

The outputs $U$ are then separated into dynamic and static components, $U_t$ and $U_{\text{static}}$, corresponding to the temporal sequence and static attributes:

$$U_t, U_{\text{static}} = U_{1:T}, U_{T+1} \tag{14}$$

Finally, both dynamic and static representations are individually projected back to their original dimensions through separate embedding layers, reconstructing the masked portions of the inputs. Dynamic variables are restored via:

$$\hat{x}_t = \text{DynamicDecEmbedding}(U_t) \tag{15}$$

while static attributes are restored by:

$$\hat{s} = \text{StaticDecEmbedding}(U_{\text{static}}) \tag{16}$$

The projections leverage the learned latent representations to reconstruct the original hydrologic inputs.

# B. Additional Experiments

## B.1. Linear Regression Baselines

To justify the use of complex neural networks over traditional methods, we have conducted baseline comparisons using linear regression models. As shown in the table below, linear regression performs poorly across all tasks by a fair margin when compared to our neural network approaches.

*Table 5.* Additional experiments with linear regression baselines.

| Experiment | Random holdout | | | Regional holdout | | |
|---|---|---|---|---|---|---|
| | RMSE ↓ | μbRMSE ↓ | Corr ↑ | RMSE ↓ | μbRMSE ↓ | Corr ↑ |
| Camels Streamflow Linear Regression | 2.190 | 2.180 | 0.500 | 2.260 | 2.250 | 0.500 |
| Soil Moisture Linear Regression | 0.120 | 0.101 | 0.188 | 0.121 | 0.103 | 0.187 |

## B.2. External Foundation Model Comparisons

For completeness, we explored several existing foundation models developed for Earth observation and atmospheric applications, including TerraMind, PrithviWxC, and Galileo (Jakubik et al., 2025; Hsu et al., 2024; Tseng et al., 2025). All models were evaluated using consistent downstream protocols, with pretrained encoders either frozen or minimally adapted and paired with task-specific heads comparable to those used for StefaLand. These experiments were intended to probe the extent to which representations learned from large-scale EO or atmospheric data transfer to land–surface and hydrologic prediction tasks.

These experiments are not intended as exhaustive benchmarks or as performance upper bounds for the evaluated models, but rather as feasibility probes to understand whether their pretrained representations can be directly repurposed for land–surface and geohazard tasks under lightweight adaptation.

Because these models differ substantially in their native input formats and pretraining objectives, task-specific adaptations were required. TerraMind and PrithviWxC were coupled with the same residual adaptation architecture used for StefaLand, with only the added adaptation units trained. Due to the intensive data and storage requirements of PrithviWxC, inputs were restricted to surface-level variables most directly related to land–surface interactions, together with static attributes, while multi-level atmospheric variables were excluded. For context, StefaLand, TerraMind, and PrithviWxC were pretrained on approximately 2, 11, and 27 terabytes of data, respectively.

*Table 6.* Performance of external foundation model baselines across tasks. All models use frozen pretrained encoders with the same residual adaptation head.

| Task | Model | RMSE ↓ | μbRMSE ↓ | Corr ↑ |
|---|---|---|---|---|
| CAMELS Streamflow (PUB) | TerraMind-resConn | 1.332 ± 0.0410 | 1.301 ± 0.0375 | 0.777 ± 0.0071 |
| CAMELS Streamflow (PUR) | TerraMind-resConn | 1.420 ± 0.2021 | 1.398 ± 0.1932 | 0.763 ± 0.0172 |
| Soil Moisture (Random) | TerraMind-resConn | 0.083 ± 0.0021 | 0.062 ± 0.0007 | 0.694 ± 0.0289 |
| | PrithviWxC-resConn | 0.081 ± 0.0019 | 0.060 ± 0.0004 | 0.703 ± 0.0390 |
| Soil Moisture (Europe) | TerraMind-resConn | 0.101 | 0.080 | 0.519 |
| | PrithviWxC-resConn | 0.103 | 0.079 | 0.523 |

## B.3. Ablations

*Table 7.* StefaLand ablations on CAMELS streamflow under random (PUB) and regional (PUR) spatial holdout.

| Variant | PUB: random holdout (ungauged basins) | | | | PUR: regional holdout (ungauged regions) | | | |
|---|---|---|---|---|---|---|---|---|
| | RMSE ↓ | ubRMSE ↓ | Corr ↑ | NSE ↑ | RMSE ↓ | ubRMSE ↓ | Corr ↑ | NSE ↑ |
| StefaLand direct | 1.882 ± 0.09 | 1.849 ± 0.07 | 0.538 ± 0.01 | 0.395 ± 0.14 | 1.982 ± 0.40 | 1.949 ± 0.39 | 0.230 ± 0.08 | 0.201 ± 1.22 |
| StefaLand scratch | 1.355 ± 0.04 | 1.332 ± 0.04 | 0.801 ± 0.00 | 0.661 ± 0.04 | 1.516 ± 0.37 | 1.378 ± 0.35 | 0.771 ± 0.02 | 0.560 ± 0.31 |
| StefaLand noResConn | 1.171 ± 0.03 | 1.154 ± 0.03 | 0.823 ± 0.00 | 0.706 ± 0.33 | 1.376 ± 0.20 | 1.356 ± 0.17 | 0.798 ± 0.02 | 0.610 ± 0.13 |
| StefaLand linear | 1.452 ± 0.05 | 1.366 ± 0.05 | 0.751 ± 0.01 | 0.661 ± 0.04 | 1.484 ± 0.26 | 1.453 ± 0.24 | 0.672 ± 0.04 | 0.542 ± 0.31 |
| resConn + Random Mask | 1.150 ± 0.11 | 1.097 ± 0.09 | 0.861 ± 0.01 | 0.720 ± 0.19 | 1.365 ± 0.38 | 1.339 ± 0.22 | 0.789 ± 0.02 | 0.629 ± 0.29 |
| resConn + No Mask | 1.210 ± 0.15 | 1.115 ± 0.11 | 0.832 ± 0.01 | 0.712 ± 0.21 | 1.386 ± 0.43 | 1.347 ± 0.27 | 0.768 ± 0.03 | 0.578 ± 0.31 |
| StefaLand resConn (CVGM) | **1.111 ± 0.04** | **1.068 ± 0.04** | **0.869 ± 0.01** | **0.717 ± 0.16** | **1.344 ± 0.21** | **1.334 ± 0.19** | **0.801 ± 0.02** | **0.635 ± 0.25** |

*Table 8.* StefaLand ablation study on soil moisture prediction under random and regional holdout.

| Variant | Random location holdout | | | Regional holdout (Europe) | | |
|---|---|---|---|---|---|---|
| | RMSE ↓ | ubRMSE ↓ | Corr ↑ | RMSE ↓ | ubRMSE ↓ | Corr ↑ |
| StefaLand direct | 0.140 ± 0.043 | 0.103 ± 0.004 | 0.637 ± 0.035 | 0.135 | 0.112 | 0.503 |
| StefaLand scratch | 0.074 ± 0.001 | 0.058 ± 0.000 | 0.749 ± 0.017 | 0.108 | 0.064 | 0.528 |
| StefaLand noResConn | 0.075 ± 0.001 | 0.057 ± 0.000 | 0.741 ± 0.020 | 0.095 | **0.058** | 0.545 |
| StefaLand linear | 0.084 ± 0.001 | 0.061 ± 0.000 | 0.720 ± 0.019 | 0.100 | 0.063 | 0.393 |
| resConn + Random Mask | 0.073 ± 0.002 | 0.059 ± 0.001 | **0.784 ± 0.007** | 0.088 | 0.061 | 0.621 |
| resConn + No Mask | 0.075 ± 0.002 | 0.062 ± 0.002 | 0.781 ± 0.006 | 0.094 | 0.063 | 0.610 |
| StefaLand resConn (CVGM) | **0.068 ± 0.001** | **0.054 ± 0.000** | 0.783 ± 0.005 | **0.090** | 0.059 | **0.638** |

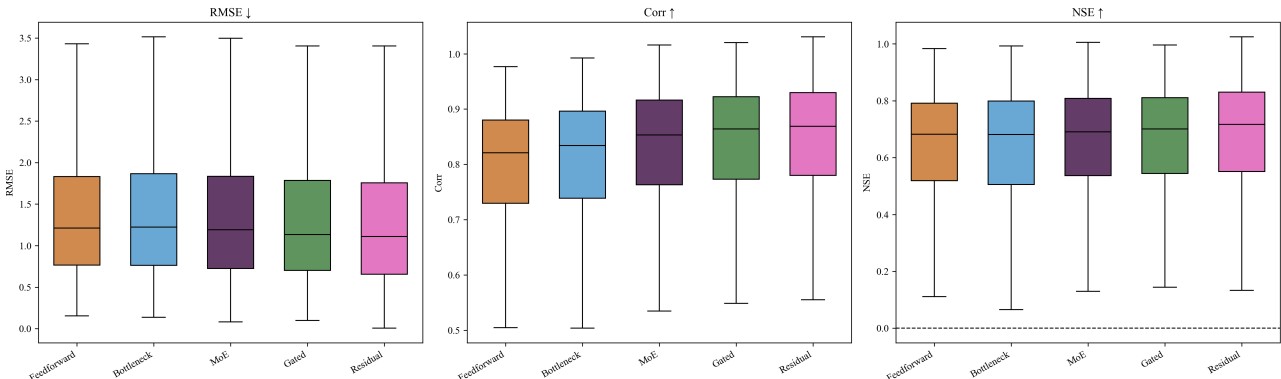

*Figure 3.* Adapter ablation on streamflow random spatial splits (PUB). Boxplots summarize per-basin performance distributions for five adapter designs (Feedforward, Bottleneck, MoE, Gated, Residual). We report RMSE (lower is better), correlation (higher is better), and NSE (higher is better) across held-out basins, showing that the Residual adapter yields the most consistent gains, particularly in Corr and NSE.

# C. Experimental Details

## C.1. Model Configurations and Hyperparameters

*Table 9.* StefaLand pretraining configuration.

| Parameter | Value |
|---|---|
| **General Settings** | |
| Task | pretrain |
| Model | Stefaland_dec_LSTM |
| Random seed | 111 |
| Time Period | 1980/1/1–2018/12/31 |
| **Sequence Configuration** | |
| Sequence length | 365 |
| Label length | 365 |
| Prediction length | 365 |
| Sampling stride | 1 |
| Minimum window size | 30 |
| Maximum window size | 90 |
| **Model Architecture** | |
| Input dimension (enc_in) | 32 |
| Decoder input (dec_in) | 6 |
| Output dimension (c_out) | 6 |
| Model dimension | 256 |
| Number of heads | 4 |
| Encoder layers | 4 |
| Decoder layers | 2 |
| Feed-forward dimension | 512 |
| Dropout | 0.1 |
| Activation | gelu |
| **Training Configuration** | |
| Optimizer | AdamW |
| Loss criterion | MaskedNSE |
| Epochs | 25 |
| Batch size | 256 |
| Learning rate | 0.0001 |
| Weight decay | 0.0 |
| Patience | 30 |
| Gradient clipping | 5.0 |
| Number of workers | 10 |
| **Loss Weights** | |
| Time series loss ratio | 1.0 |
| Static loss ratio | 0.5 |

*Table 10.* Attribute groups used in group masking pretraining.

| Group | Variables |
|---|---|
| **Topography** | meanelevation, meanslope |
| **Soil** | HWSD_clay, HWSD_sand, HWSD_silt, HWSD_gravel, SoilGrids1km_sand, SoilGrids1km_clay, SoilGrids1km_silt |
| **Geology** | permeability, Porosity, glaciers, permafrost |
| **Vegetation** | NDVI, FW |
| **Climate** | aridity, meanP, ETPOT_Hargr, meanTa, seasonality_P, seasonality_PET, snow_fraction, snowfall_fraction |

*Table 11.* Soil moisture model configuration.

| Parameter | Value |
|---|---|
| **General Settings** | |
| Mode | traintest |
| Random seed | 111111 |
| Data loader | onlylstmloader |
| Data sampler | finetuningnoHBV |
| **Training Configuration** | |
| Time period | 2015/04/01–2020/12/31 |
| Target | soil_moisture |
| Optimizer | Adadelta |
| Batch size | 128 |
| Epochs | 50 |
| Save frequency | Every 25 epochs |
| **Neural Network Configuration** | |
| Hidden size | 128 |
| Dropout | 0.3 |
| Learning rate | 1.2 |
| Encoder layers | 16 |
| Decoder layers | 12 |
| Feed-forward dimension | 512 |
| Rho | 365 |
| **Loss Function** | |
| Type | RmseLoss |

## C.2. Variables and Data Sources

*Table 12.* StefaLand pretraining variables and sources.

| Variable Type | Variable Name | Source |
|---|---|---|
| **Time Series Forcings** | Precipitation, Short-wave solar radiation downwards, Relative humidity, Maximum temperature, Minimum temperature, Potential evapotranspiration | from Multi-Source Weather (MSWX) and Multi-Source Weighted-Ensemble Precipitation (MSWEP) (Beck et al., 2022; 2019) |
| **Static Attributes** | Forest cover fraction, grassland cover fraction | Climate Change Initiative (CCI) land cover dataset (ESA, 2017) |
| | Normalized Difference Vegetation Index (NDVI) | Terra Moderate Resolution Imaging Spectroradiometer (MODIS) Vegetation Indices (MOD13A3) (Didan, 2015) |
| | Sand, silt, clay fractions | Harmonized World Soil Database (HWSD) (FAO et al., 2012) |
| | Elevation, slope, aspect | Global Multi-resolution Terrain Elevation Data (GMTED) (Danielson & Gesch, 2011; Ramcharan et al., 2018) |
| | Soil depth | Global 1-km Gridded Thickness of Soil, Regolith, and Sedimentary Deposit Layers (Pelletier et al., 2016) |
| | Carbonate sedimentary rock fraction | Global Lithological Map (GLiM) (Hartmann & Moosdorf, 2012) |
| | Rock porosity, permeability | GLobal HYdrogeology MaPS (GLHYMPS) (Gleeson et al., 2014) |
| | Population density | Gridded Population of the World (GPW) v4 dataset (CIESIN, 2016) |
| | GDP per capita; population density | Gross Domestic Product and Human Development Index over 1990-2015 (Kummu et al., 2018) |
| | Forest intact fraction | Intact Forest Landscapes Data (Potapov et al., 2017) |
| **Outputs** | None (self-supervised pretraining) | — |

*Table 13.* CAMELS streamflow variables and sources.

| Variable Type | Variable Name | Source |
|---|---|---|
| **Time Series Forcings** | Precipitation, Temperature, Potential evapotranspiration, Solar radiation, Vapor pressure | Catchment Attributes and Meteorology for Large-sample Studies (CAMELS) (Addor et al., 2017; Newman et al., 2014) |
| **Static Attributes** | Elevation, slope, catchment area, forest cover, LAI, GVF, soil depth, porosity, conductivity, sand, silt, clay fractions, carbonate fraction, permeability, aridity, snow fraction, precipitation extremes | CAMELS |
| **Outputs** | Streamflow | CAMELS gauge records |

*Table 14.* Soil moisture variables and sources.

| Variable Type | Variable Name | Source |
|---|---|---|
| **Time Series Forcings** | Albedo (BSA, WSA) | Moderate Resolution Imaging Spectroradiometer (MODIS) MCD43A3 version 6 (Schaaf, Crystal & Wang, Zhuosen, 2021) |
| | LST (Day, Night) | MODIS Land Surface Temperature/Emissivity Daily (MYD11A1) Version 6.1 (Wan et al., 2021) |
| | Precipitation | Global Precipitation Measurement (GPM), MSWEP, and ERA5 precipitation (Huffman et al., 2019; Beck et al., 2019; Muñoz Sabater, 2019) |
| | Forecast albedo, LAI (high/low vegetation), soil temperature (layer 1), surface pressure, solar radiation, 2 m temperature, evaporation, precipitation, U/V wind (10 m) | ECMWF Reanalysis v5 (ERA5) (Muñoz Sabater, 2019) |
| **Static Attributes** | elevation, slope, aspect, roughness, curvature | Global 1/5/10/100-km topography derivatives (Amatulli et al., 2018) |
| | Sand, clay, silt, bulk density | HWSD v1.2 (FAO et al., 2012) |
| | Land cover; urban; open water; snow/ice | ESA CCI Land Cover (ESA, 2017) |
| | NDVI | Vegetation Indices Monthly L3 Global 0.05Deg CMG (Didan et al., 2015) |
| **Outputs** | Soil moisture | International Soil Moisture Network (ISMN) (Dorigo et al., 2013b; 2011) |

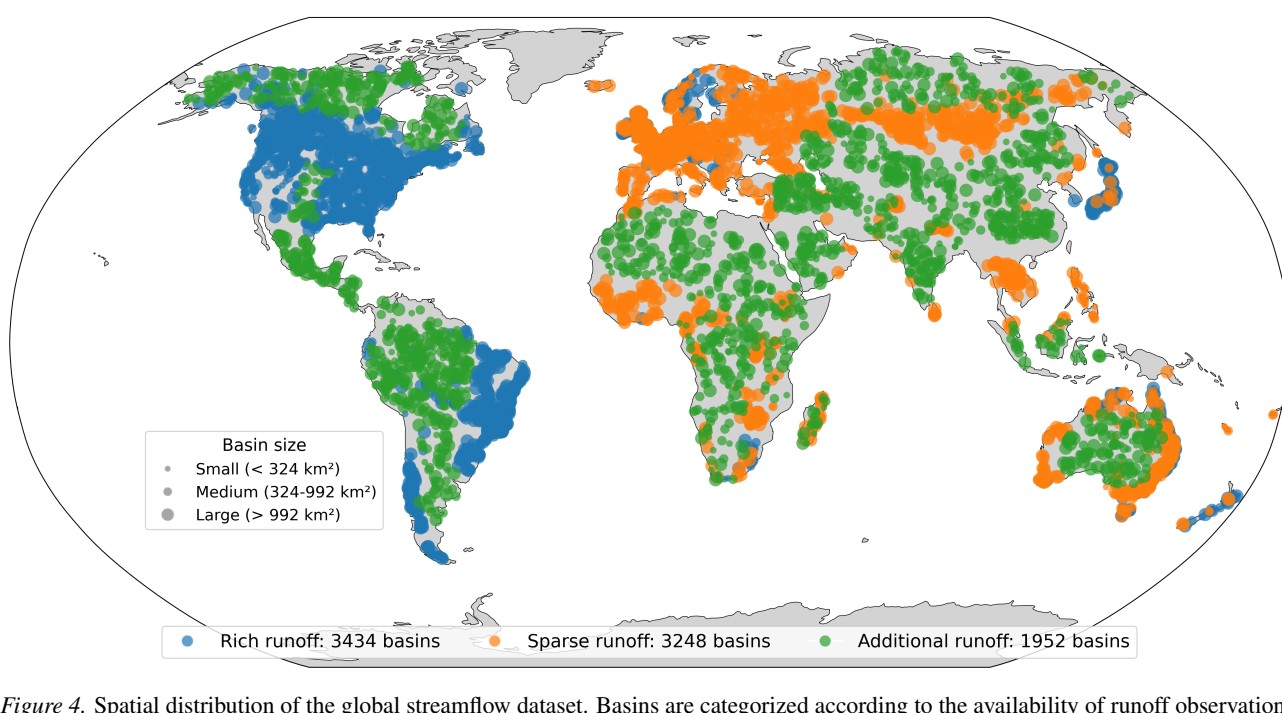

*Figure 4.* Spatial distribution of the global streamflow dataset. Basins are categorized according to the availability of runoff observations: basins with relatively abundant runoff records (blue), basins with sparse runoff records (orange), and basins without runoff data (green). Marker size corresponds to basin area, classified into three categories based on the 33rd and 67th percentiles of catchment areas.

## C.3. Computational Resources

*Table 15.* Computational resources for StefaLand and comparison models.

| Task | Model | Dataset | Sec/Epoch | #GPUs | GPU | GPU hrs* |
|------|-------|---------|-----------|-------|-----|----------|
| Pretraining | StefaLand | Global | 16,000 | 6 | V100 | ∼720 |
| Finetuning | ResConn | CAMELS | 30 | 2 | V100 | ∼4 |
| | no Adapter | CAMELS | 26 | 2 | V100 | ∼3.6 |
| | ResConn | Soil Moisture | 43 | 2 | V100 | ∼2.9 |
| | ResConn | Soil Comp. | 8 | 2 | V100 | ∼0.6 |
| | ResConn | Landslide | 3,542 | 2 | RTX 2080Ti | ∼1 |
| Baseline | LSTM | CAMELS | 12 | 2 | V100 | ∼1.6 |
| | LSTM | Soil Moisture | 21 | 2 | V100 | ∼1.45 |
| | Random Forest | Soil Comp. | 5 | 2 | V100 | ∼0.3 |
| | CNN2D | Landslide | 370 | 2 | RTX 2080Ti | ∼0.1 |

*Total GPU hours reflect the random holdout scenario where applicable. All StefaLand finetuning rows use the pretrained StefaLand encoder as the base model.

## C.4. Pretraining Data Handling

A global dataset was constructed for model pretraining, including 8,634 catchments and designed to characterize climatic, ecological, soil, topographic, geological, and socioeconomic conditions. The dataset includes both daily meteorological forcings and long-term averaged static attributes. Daily meteorological variables comprise precipitation, downward shortwave radiation, relative humidity, maximum temperature, and minimum temperature, derived from the Multi-Source Weather (MSWX) and Multi-Source Weighted-Ensemble Precipitation (MSWEP) datasets at a spatial resolution of 0.1° (Beck et al., 2022; 2019). Potential evapotranspiration was estimated using the Hargreaves method.

The study catchments are divided into three groups: 3,434 GRDC catchments with relatively abundant historical runoff records, 3,248 GRDC catchments with sparse runoff records, and 1,952 HydroBasins level-8 catchments without runoff observations (GRDC, 2024; Lehner & Grill, 2013). Ecosystem states are represented by forest and grassland cover fractions

derived from the Climate Change Initiative (CCI) land cover dataset (ESA, 2017), along with the Normalized Difference Vegetation Index (NDVI) from MODIS (Didan, 2015). Soil properties include sand, silt, and clay fractions from the Harmonized World Soil Database (HWSD) (FAO et al., 2012).

Topographic attributes include elevation, slope, and aspect obtained from the Global Multi-resolution Terrain Elevation Data (GMTED) (Danielson & Gesch, 2011; Ramcharan et al., 2018), as well as terrain-derived soil depth from the Global 1-km Gridded Thickness of Soil, Regolith, and Sedimentary Deposit Layers dataset (Pelletier et al., 2016). Geological attributes comprise carbonate sedimentary rock fractions from the Global Lithological Map (GLiM) (Hartmann & Moosdorf, 2012) and rock porosity and permeability from the GLobal HYdrogeology MaPS (GLHYMPS) dataset (Gleeson et al., 2014).

Socioeconomic conditions are characterized using population density from the Gridded Population of the World (GPW) v4 dataset (CIESIN, 2016), gross domestic product and population data from the gridded global GDP and Human Development Index datasets (Kummu et al., 2018), and forest intactness from the Intact Forest Landscapes dataset (Potapov et al., 2017). All static attributes were mapped to a common 0.01° grid prior to basin-scale aggregation to ensure spatial consistency across data sources and improve spatial averaging over irregular basin geometries. A complete list of variables is provided in Table 12.

**C.5. Dataset Splitting**

For the WoSIS soil dataset, we collected soil property data from 106,503 locations. After removing low-quality records (e.g., sand values greater than 1 or negative values), we randomly sampled 5,000 soil points to reduce computational cost. We then applied 5-fold cross-validation (k=5) on this subset.

For the landslide dataset at 30 m resolution, we used 14,604 historical landslide points. We split the dataset into 70% for training, 20% for validation, and 10% for testing.

## D. Additional Figures

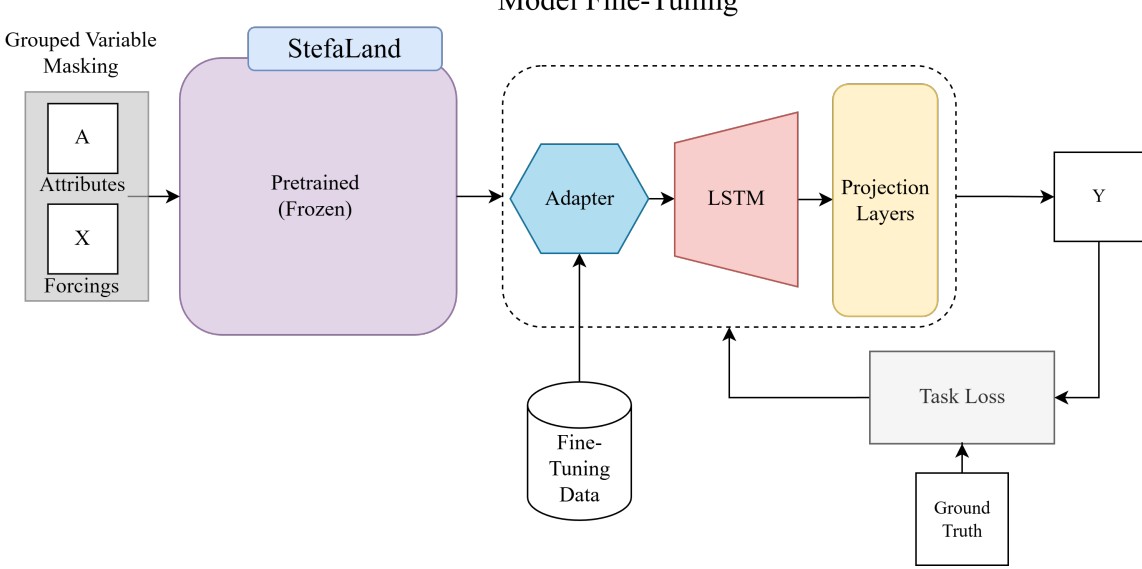

*Figure 5.* Fine-tuning pipeline used in our downstream experiments. Static attributes and meteorological forcings are encoded by the pretrained StefaLand encoder (frozen), then passed through a task adapter and sequence model (LSTM), followed by projection layers to generate predictions optimized with a task loss against ground truth.

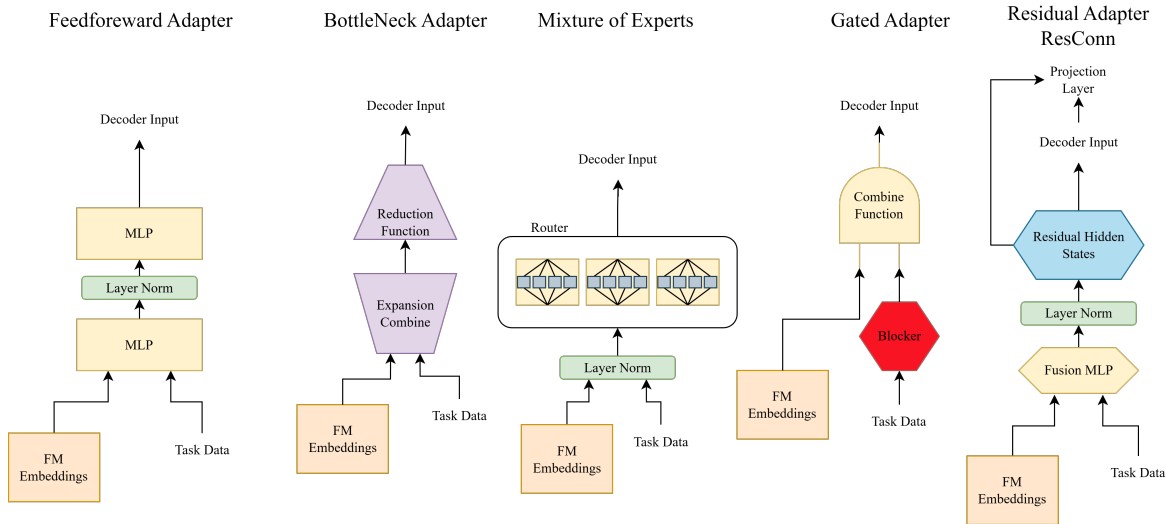

*Figure 6.* Adapter architectures evaluated in our experiments. We compare a gated adapter, a bottleneck adapter with compression and expansion stages, A mixture of Experts and a basic feedforeward and a residual adapter that injects pretrained features via a skip connection.

# E. Metric Calculations

This appendix details the calculation of the evaluation metrics used in our experiments. All metrics presented in the main paper tables are the median values across test basins or stations, as computed using the following formulations.

## E.1. Primary Evaluation Metrics

### E.1.1. ROOT MEAN SQUARE ERROR (RMSE)

RMSE measures the average magnitude of prediction errors. Lower values indicate better performance.

$$\text{RMSE} = \sqrt{\frac{1}{n}\sum_{i=1}^{n}(y_{\text{pred},i} - y_{\text{target},i})^2} \tag{17}$$

### E.1.2. UNBIASED ROOT MEAN SQUARE ERROR (µBRMSE)

µbRMSE removes the bias component from the error calculation, focusing on the error's random component. It is calculated by first computing anomalies from the mean for both predictions and targets.

$$y'_{\text{pred},i} = y_{\text{pred},i} - \overline{y}_{\text{pred}} \tag{18}$$

$$y'_{\text{target},i} = y_{\text{target},i} - \overline{y}_{\text{target}} \tag{19}$$

$$\text{µbRMSE} = \sqrt{\frac{1}{n}\sum_{i=1}^{n}(y'_{\text{pred},i} - y'_{\text{target},i})^2} \tag{20}$$

### E.1.3. CORRELATION (CORR)

Correlation quantifies the linear relationship between predictions and targets. Values range from -1 to 1, with 1 indicating perfect positive correlation.

$$\text{Corr} = \frac{\sum_{i=1}^{n}(y_{\text{pred},i} - \overline{y}_{\text{pred}})(y_{\text{target},i} - \overline{y}_{\text{target}})}{\sqrt{\sum_{i=1}^{n}(y_{\text{pred},i} - \overline{y}_{\text{pred}})^2 \sum_{i=1}^{n}(y_{\text{target},i} - \overline{y}_{\text{target}})^2}} \tag{21}$$

This is calculated using Pearson's correlation coefficient between predicted and observed values.

## E.2. Secondary Metrics

The following metrics are used in our comprehensive evaluation but may not appear directly in the main tables.

### E.2.1. NASH-SUTCLIFFE EFFICIENCY (NSE) / $R^2$

NSE evaluates the predictive skill relative to using the mean of observations as a predictor. Values range from $-\infty$ to 1, with 1 indicating perfect prediction.

$$\text{NSE} = 1 - \frac{\sum_{i=1}^{n}(y_{\text{target},i} - y_{\text{pred},i})^2}{\sum_{i=1}^{n}(y_{\text{target},i} - \overline{y}_{\text{target}})^2} \tag{22}$$

### E.2.2. MEAN ABSOLUTE ERROR (MAE)

MAE measures the average absolute difference between predictions and targets.

$$\text{MAE} = \frac{1}{n}\sum_{i=1}^{n}|y_{\text{pred},i} - y_{\text{target},i}| \tag{23}$$

### E.3. Metric Aggregation

For each evaluation scenario (Random Holdout and Regional Holdout), metrics were calculated for each individual basin or station and then aggregated using median values to provide a robust measure of central tendency less sensitive to outliers. All metrics shown in tables throughout the paper represent these median values across the test set.

### E.4. Implementation Details

All metrics were implemented in Python using NumPy for numerical computations and SciPy's statistical functions for correlation coefficients. Special care was taken to handle missing values (NaNs) appropriately in all calculations. For time series with missing values, only timestamps where both predicted and target values were available were used in metric calculations.

