# OpenReview forum: "Structured Physical Attributes Enable Efficient Foundation Models for Land-Surface Prediction"
_ICML.cc/2026/Workshop/FMSD — FMSD @ ICML 2026 Poster_

### Official Review · Reviewer_N9fR · 2026-05-22
**Review of the paper - Structured Physical Attributes Enable Efficient Foundation Models for Land-Surface Prediction**

**Rating:** 4
**Confidence:** 3

**Review:**

**Summary** - This paper introduces StefaLand (Spatial-Temporal Earth Foundation model with Attributes for the Land Surface), a transformer-based masked autoencoder pretrained on tabular landscape attributes and meteorological time-series. The model's primary contribution is Cross-Variable Group Masking (CVGM), in which physically related attribute groups (topography, soil, geology, etc) are masked jointly during pretraining, forcing the encoder to reconstruct cross-domain interactions rather than exploiting within-group correlations. Evaluated on two targets - streamflow forecasting (CAMELS) and soil moisture estimation (ISMN), StefaLand-resConn generally outperforms baselines on streamflow and achieves the strongest correlation on soil moisture, although its soil moisture rmse/ubRMSE gains are mixed under cross-continental holdout.

**Strengths**

1. The problem statement is clearly motivated - general-purpose tabular/TS foundation models lack physical inductive biases and vision-based EO foundation models encode spatial patterns rather than structured attribute interactions.

2. The CVGM strategy is interesting and meaningful. Masking by physical domain compels the model to internalize known geophysical co-dependencies.

3. Strong empirical evaluation. The paper follow established community benchmarking protocols. Moreover, the reported gains are substantial, for e.g. approx. 20% RMSE reduction over LSTM-SL in PUB, and around 16% in PUR. Additionally, useful ablation studies have been carried out.

**Weaknesses**

1. There geographic diversity of evaluation benchmarks despite global pretraining, is limited. For instance, the streamflow evaluation is entirely restricted to CAMELS-US. Despite the pretraining dataset spanning 8k+ global catchments including data-scarce Africa, South America, and Asia, no downstream evaluation is conducted in these regions. This is a concern given these are exactly the regions motivated in Section 1 as the use case.

2. The CVGM ablation benefit seems modest and inconsistently reported. Removing CVGM in favour of random masking increases rmse by only around 3% (CAMELS regional), and by -2% for ISMN regional respectively. Table 7 supports this - resConn + random mask achieves rmse 1.150 vs. full model 1.111 on PUB, a difference of only 0.039 that may not be statistically significant given the standard errors reported. The claim that "physically motivated cross-variable grouping contributes to cross-domain representation learning" is not fully supported by these results.

3. Similarly, soil moisture cross-continental results show mixed and inconsistently favourable patterns. In Europe holdout, StefaLand-resConn achieves the best correlation, but its RMSE is worse than TabPFN, Reformer, Informer, and AlphaEarth-LSTM, and only tied with AlphaEarth-resConn. It also does not win on ubRMSE; LSTM-SL, DLinear, TabPFN, and Reformer are better. The discussion selectively emphasizes correlation, but for operational geohazard warning, RMSE, ubRMSE, and bias-sensitive metrics may be equally or more important.

4. The design choices are unclear. Wouldn't the summation of per-variable dynamic embeddings across the channel dimension destroy per-variable identity before the Transformer? It would potentially limit the expressiveness of cross-variable attention. Some ablations and analysis would be helpful.

5. The pretraining uses a bidirectional LSTM reconstruction decoder on top of the Transformer encoder, adding more complexity without a clear ablation of how it outperforms a simple linear projection decoder (which is very common in TS modeling).

6. There is no physics-informed hydrology baseline. There are multiple differentiable physics-informed LSTM hybrids developed by the hydrological modeling community. Given that the paper's central claim is about physically meaningful pretraining for hydrological generalization, it should compare against at least one differentiable/process-guided hydrology model or explain why such models are not comparable under the chosen protocol.

---

### Official Review · Reviewer_kQmp · 2026-05-22
**Promising domain-specific Foundation Model for land-surface prediction, but claims need clearer validation and reproducibility details**

**Rating:** 6
**Confidence:** 4

**Review:**

**Summary**

This paper proposes StefaLand, a structured foundation model for land-surface prediction. The model is pretrained on static landscape attributes and meteorological time series across 8,634 global catchments using a transformer-based masked autoencoder with cross-variable group masking. The authors adapt the pretrained encoder to downstream tasks using a lightweight residual adapter and evaluate it on two Earth system prediction tasks: streamflow forecasting and soil moisture estimation.

The paper reports that StefaLand improves spatial generalization under strict holdout settings. On CAMELS streamflow prediction, StefaLand-resConn achieves the best performance in both ungauged basin and ungauged region settings, reducing RMSE relative to supervised LSTM baselines. On soil moisture prediction, StefaLand also achieves strong correlation and competitive or best RMSE across random and cross-continental holdout settings.

**Strengths**

1. The paper addresses an important problem: improving spatial generalization for land-surface prediction in data-sparse regions.

2. The proposed foundation model performs well on both streamflow and soil moisture tasks, especially under spatial holdout settings.

**Areas for Improvement**

1. The paper makes strong claims about foundation model transfer, but the evaluation is still limited to two downstream tasks. Broader evaluation on additional land-surface variables, such as evapotranspiration or water quality, would better support the claim.

2. The comparison to general-purpose foundation models may not be fully fair. TabPFN and TiRex are not naturally designed for this type of spatially structured, physically governed, multivariate land-surface prediction. The paper should more carefully distinguish between "general-purpose foundation models are weak for this task'' and "the adaptation protocol used here is not optimal for them.''

3. The contribution of pretraining versus architecture needs clearer isolation. The ablation figure suggests that pretraining is important, but some variants such as random masking and no masking are relatively close to the full model in certain settings. The paper should more carefully quantify when cross-variable group masking provides substantial gains.

4. Some results have large standard errors, especially in regional holdout settings. The paper should discuss whether the improvements are statistically significant across folds and regions.

---

### Official Review · Reviewer_gpeH · 2026-05-22
**Masked-autoencoder pretraining on physical variables for hydrologic prediction with downstream finetuning**

**Rating:** 5
**Confidence:** 3

**Review:**

# Summary
StefaLand is a transformer-based masked autoencoder pretrained on lumped catchment attributes and meteorological time series across 8,634 global catchments. It introduces cross-variable group masking (CVGM) and evaluates on streamflow (CAMELS) and soil moisture (ISMN) under spatial holdout. Results show improvements over supervised LSTMs and general-purpose foundation models (TabPFN, TiRex).

# Strengths

- A worthwhile finding for this workshop: general-purpose structured FMs fail on physically-governed long memory systems.
- Thorough ablations isolating pretraining, adaptation, and CVGM.
- Well written with sufficient experiments

# Areas for Improvement

- The "foundation model" framing is not well-supported. The model is always finetuned (adapter + LSTM decoder trained on labeled data) before evaluation. No zero-shot or in-context evaluation. This is a pretrain-then-finetune pipeline, not a foundation model.

- Pretraining is standard self-supervised training on a single domain. The corpus is exclusively hydro-meteorological forcings and static attributes. Calling this a foundation model conflates domain-specific pretraining with the generality expected of FMs.

- The downstream task is nowcasting, not forecasting. The model gets concurrent forcings and predicts the response at the same timestep, this is the easiest possible setup for neural hydrology. There is a large literature on how to improve this setup, yet the paper benchmarks only against the initial 2019-era LSTM approach (Kratzert et al., 2019). More importantly, the real operational and generalization challenge is multi-step-ahead forecasting with weather forecasts. To make a credible case, the paper should adopt the forecasting setup used by Google's encoder–decoder LSTM (Nearing et al., 2024) and RiverMamba. The nowcasting-only choice limits practical relevance and makes the FM label harder to justify.

- The compute argument is weak given the standard hydrology baseline. The dominant methodology in hydrology is already a lightweight LSTM that can be trained efficiently worldwide in a couple of hours on a single V100 (Nearing et al., 2024). Against this baseline, a 720 V100-hour pretraining stage plus per-task finetuning is not obviously "computationally accessible," and the efficiency claim should be benchmarked against the LSTM rather than only against vision-based earth models.

- Fit to workshop scope is questionable. The workshop emphasizes zero-shot/ICL FMs (TabPFN, TiRex, Chronos, TimesFM). StefaLand always requires finetuning and demonstrates no cross-domain transfer.

Questions:
1. Can you provide any zero-shot or linear-probe-only evaluation?
2. What prevents this architecture from performing multi-step-ahead forecasting with weather forecasts?
3. How does total compute (pretraining + finetuning) compare to simply training a worldwide LSTM, which takes only a couple of hours on one V100?
4. How sensitive are results to the CVGM group definitions?

# Justification of Score

A weak accept is warranted for the application relevance, the insight that general-purpose structured FMs fail on physically-governed systems, and the careful evaluation protocol. However, the foundation model framing is not justified, the model is a standard self-supervised pretrained encoder on a single domain that always requires finetuning, with no zero-shot/ICL capability; only nowcasting (the easiest neural-hydrology setup) is evaluated rather than true forecasting; the efficiency claim is unconvincing against the lightweight LSTM that already dominates hydrology; and the fit to this workshop's scope is tenuous.